# The Use of Chest Magnetic Resonance Imaging in Malignant Pleural Mesothelioma Diagnosis

**DOI:** 10.3390/diagnostics12030750

**Published:** 2022-03-19

**Authors:** Federica Volpi, Caterina A. D’Amore, Leonardo Colligiani, Alessio Milazzo, Silvia Cavaliere, Annalisa De Liperi, Emanuele Neri, Chiara Romei

**Affiliations:** 1Department of Translational Research, Academic Radiology, University of Pisa, 56126 Pisa, Italy; f.volpi2@studenti.unipi.it (F.V.); c.damore@studenti.unipi.it (C.A.D.); l.colligiani@studenti.unipi.it (L.C.); a.milazzo4@studenti.unipi.it (A.M.); emanuele.neri@unipi.it (E.N.); 2Department of Diagnostic Imaging, Diagnostic Radiology 2, Pisa University Hospital, 56124 Pisa, Italy; s.cavaliere@ao-pisa.toscana.it (S.C.); a.deliperi@ao-pisa.toscana.it (A.D.L.)

**Keywords:** magnetic resonance imaging, malignant pleural mesothelioma, malignant pleural disease

## Abstract

In recent years, many articles have demonstrated that magnetic resonance imaging (MRI) may be performed successfully in the study of the chest. The aim of this study was to evaluate the potential role of MRI in the differentiation of benign from malignant pleural disease with a special focus on malignant pleural mesothelioma and on MRI protocols. A systematic literature search was performed to find original articles about chest MRI in patients with either benign or malignant pleural disease. We retrieved 1246 papers and 17 studies were finally identified as being in accordance with our purpose. For a morphologic assessment, T1-weighted and T2-weighted sequences were usually performed, eventually associated with T1 post-contrast sequences for better detection of pleural lesions. Functional sequences such as Diffusion Weighting Imaging (DWI), associated with the evaluation of Apparent Diffusion Coefficient (ADC) maps, were lately and gradually introduced in chest MRI protocols and their potentiality in differentiating benign from malignant disease has been investigated by many authors. Many progresses have been performed to improve quality images and diagnostic performances of MRI. A better and early identification of pleural disease may be obtained, providing MRI as a possible tool that can differentiate malignant from benign pleural disease without using invasive procedures.

## 1. Introduction

Malignant Pleural Mesothelioma (MPM) is a rare aggressive tumor that derives from mesothelial cells found in the pleural and peritoneal surface [1]. MPM is the most common primary pleural neoplasm, although metastases are still the most frequent pleural malignant lesions [2]. MPM is strongly related to asbestos exposure in 40–80% of patients, with a latency period of 45 years [2]. In addition, previous chest radiotherapies (for lymphoma, breast or lung cancers), simian virus 40 infection (SV40) and oncogenic mutations such as the Breast Cancer gene 1-associated protein (BAP 1) were highlighted as potential risk factors for MPM [2,3,4]. Male patients are most affected (men to women ratio of 4:1) with a median age among 50 and 70 years [2,5].

Patients with MPM are often symptomatic with dyspnea and chest pain; other symptoms may be cough, malaise and weight loss [2]. The presence of monolateral pleural effusion is a frequent sign (79%). Mediastinal lymph nodes and involvement of surrounding structures may cause dysphagia, superior vena cava syndrome, phrenic nerve paralysis, and cardiac tamponade [2,5].

MPM has a poor prognosis with a median survival of 4 to 18 months [2]. Nowadays treatment options include chemotherapy, targeted therapy and immunotherapy, radiotherapy and surgery [2,5,6]. Poor performance status, male gender, non-epithelioid subtype, low hemoglobin, high platelet and leucocytes count, and increased lactate dehydrogenase, were highlighted as predictors of worst outcome [2,4].

MPM diagnosis is still based on histological assessment through thoracoscopy or percutaneous biopsies with imaging guidance by computed tomography (CT) or ultrasound (US) [2]. In the past years, several studies tried to identify potential diagnostic biomarkers, and among them only mesothelin seems promising. Mesothelin is a cell-surface glycoprotein overexpressed in MPM and identified on serum and pleural samples [5,7].

Imaging examinations are crucial in MPM diagnosis and staging; in particular, chest radiography is the first-line test to assess pleural lesion, although it has a low sensitivity and specificity. Pleural effusion, pleural masses or plaques may be detectable on an X-ray but CT results as the gold standard for pleural lesions evaluation [8]. CT allows a better depiction of pleural lesions and may help in the differentiation between benign and malignant alterations. Typically, the presence of a pleural thickness higher than 1 cm, irregular or nodular pleural thickening, and mediastinal pleural involvement with/without extension to the diaphragm and chest wall are suggestive of malignancy, together with a high enhancement after intravenous contrast administration [4]. The possible differential diagnoses are pleural metastasis, pleural solitary fibrous tumor, epithelioid hemangioendothelioma, and pleural involvement of aggressive thymoma [2,8]. However, CT limits are well known as it determines radiation exposure and the administration of iodine contrast, which is affected by more frequent adverse reaction, particularly in patients with diabetes, chronic renal failure and previous allergy [9]. In addition, CT evaluation for locoregional staging presents a high interobserver variability [10]. Positron emission tomography (PET)-CT with FDG is indicated for systemic staging. A standardized uptake value (SUV) threshold of two was identified as useful in the differential diagnosis between benign and malignant disease [5,8]. Moreover, PET/CT helps in biopsy planning. False positive lesions on PET/CT may be found in the case of previous pleurodesis, tuberculosis or inflammatory disease [8]. Recently, combined PET/MRI imaging techniques were investigated too, in order to integrate the high soft tissue contrast of MRI to the functional analysis of PET, providing an accuracy comparable to that of PET/CT in MPM staging and a higher diagnostic confidence [11].

The role of MRI in pleural disease evaluation and in the diagnosis of MPM has long been discussed over the past 20 years. In particular, in the last 10 years, the number of reviews and articles regarding the role of imaging in MPM assessment increased. Moreover, international conferences, such as the International Conference of the International Mesothelioma Interest Group (iMig), founded in 1991, analyzed the role of imaging in the management, diagnosis, prognosis and response to therapies of MPM [12,13,14].

MRI was found to be superior to CT in the evaluation of pleural lesions, in the presence of small foci, and in the differentiation of malignant from benign disease [15,16]. Due to the high soft tissues contrast and spatial resolution, MRI provides the best evaluation of loco-regional involvement resulting in an important tool for MPM staging and treatment management [5,8,15,16]. MRI is important to investigate the resectability of pleural tumors and to plan surgery in selected patients [17,18].

MRI limits are well known, such as the higher cost and the reduced availability of scans. Moreover, MRI acquisition takes at least 20–25 min, and this could be a limit in patients with low performance status and breathing difficulties [19]. Regardless, the advantages of MRI are the lack of radiation exposure, the less frequent contrast media reactions, and lower nephrotoxicity [16,20] in addition to the possibility of quantitative imaging analysis based on Diffusion-weighted imaging (DWI) and dynamic contrast-enhanced (DCE) acquisition. This could be important for lesion characterization and tumor response evaluation after treatment [8].

The aim of this study was to evaluate the potential role of MRI in the differentiation of benign from malignant pleural disease with a special focus on malignant pleural mesothelioma and on MRI protocols.

## 2. Materials and Methods

A systematic literature research was performed to identify all relevant data on the potential role of MRI in differentiating benign from malignant pleural disease.

### Search Strategies

The databases utilized for the research of relevant publications were Scopus, Web of Science and PubMed. The last search was run on 20 November 2021. The keywords used to identify articles included benign pleural disease; malignant pleural disease; malignant pleural mesothelioma; magnetic resonance imaging; MRI. Filters were applied to include only articles published in English, described as original research and published after the year 2000. Only articles with a sample size greater than 10 patients were included.

Two authors (AM and FV) independently performed a first selection of the articles based on the title. Subsequently, the abstracts of the identified studies were screened and the full text of studies that passed the title and abstract screening was read. Any disagreement was overcome by discussion and reaching a mutual agreement.

## 3. Results

A total of 1246 studies were obtained from the research conducted. In total, 1090 records were removed for lack of inherence, while there were 112 duplicates. Titles and abstracts of the remaining 44 records were screened and after this process 35 articles were assessed for eligibility. Each article was fully read and after this process 18 papers were excluded because they did not meet all eligibility criteria. Finally, 17 studies were included in this review (Figure 1), (Table 1).

## 4. Discussion

### 4.1. Morphological Evaluation

When evaluating pleural lesions on MRI, a morphological assessment should be performed primarily. To achieve proper anatomical imaging and to reduce susceptibility artifacts, the Turbo Spin Echo (TSE) T2w sequence with a short echo spacing sequence, on axial, coronal and sagittal plans is recommended [9,15,17,22]. TSE T2w sequences with fat suppression are also suggested [15]. To reduce motion artifacts, breath-hold acquisition and cardiac triggering should be considered. In clinical practice many sequences are obtained with breath-hold techniques in inspiration, but the application depends on patient compliance [9]. Respiratory gating could be an alternative [16].

Malignant pleural mesothelioma shows an inhomogeneous hypointense to isointense signal on T1w and hyperintense signal on T2w MR sequences (Figure 2) and enhancement after contrast administration [9,16]. Mediastinal pleural involvement, nodular and/or circumferential pleural thickening (more than 1 cm), nodularity and/or infiltration of adjacent structures as chest wall or diaphragm are suggestive of malignant pleural disease [16,23,24]. Typically, a retraction of the involved hemithorax is detected, also called by Hierholzer et al., “shrinking lung” [24,25].

In 2021, Sabri et al., highlighted MRI morphological features such as contour and thickness as potential tools to differentiate malignant from benign lesions with a sensitivity of 89.29%, specificity of 76%, positive predictive value of 89%, negative predictive value of 76.92% and accuracy of 85.37% [26].

In 2004, Weber et al. evaluated the role of a very short T2* gradient echo pulse sequence with fat suppression and radial K-space acquisition. The MRI protocol also included high-resolution breath-hold T1w TSE, a respiratory-gated T2w TSE, half-Fourier single-shot turbo spin-echo (HASTE) sequences and finally a contrast-enhanced T1w acquisition with fat suppression. Radial MRI application provides a shorter acquisition time because of its ultrashort echo time of 0.5 milliseconds. Due to the repeated sampling of the k-space, it is helpful in reducing the effect of motion artifact due to cardiac and breath activity [16].

In a study by Plathow et al., HASTE sequences were defined as optimal for tumor delineation and pleural fluid evaluation [19]. In this study, in addition to the mentioned morphological features, volumetric analysis was proposed to assess tumor and lung volumes. For this purpose, first an automatic segmentation was attempted, then a semiautomated segmentation was preferred as the automatic one was affected by signal intensity inhomogeneities and motion artifacts. Patients also underwent spirometry. The volumetric evaluation showed a good correlation with treatment response and spirometry [19].

### 4.2. Functional Imaging

CT is known to be a feasible and fast imaging modality for the assessment of structural changes in many chest pathologies, including pleural disease. The development of new MRI techniques, such as functional ones, has also led MRI to gain more importance in the study of the chest, where the use of contrast agent has proven to be useful for a better pleural evaluation [17,22], therapeutical planning and for the assessment of tumor response [18,19].

Diffusion weighted imaging may be useful for a better detection of pleural disease, helping in the differentiation of a benign origin of the disease from a malignant one and to distinguish malignant lesions from each other, also thanks to the involvement of post-processing analysis (ADC maps) [27,28].

A concordance on a threshold value for mean ADC able to differentiate lesions is still missing, although different technique of region of interest (ROI) positioning has been investigated to reduce errors in the collection of these values [29,30].

An easy, fast way of recognizing of malignant pleural lesions was also assessed and proposed, as to avoid the use of ADC [25].

#### 4.2.1. Dynamic MRI

In 2006, Plathow et al., combined morphological and functional evaluation comparing standard 2D MRI and novel 3D dMRI (dynamic MRI) techniques. Functional MRI is of great importance not only in surgical/therapeutical planning but even in the assessment of tumor response. Two-dimensional dMRI was performed using true-FISP (time-resolved true fast imaging with steady-state precession) sequence measuring the displacement of chest wall and diaphragm on a workstation (SYNGO, Siemens Medical Solutions, Erlangen, Germany). Three-dimensional dMRI was obtained recording lung movement during breathing cycles in three dimensions (3D) using an isotropic time-resolved 3D gradient echo pulse sequence associated with the view-sharing implementation TRICKS (time-resolved interpolated contrast kinetics). Three-dimensional dMRI was found to provide functional parameters indicative of therapeutic response not available with simple spirometry. Regardless, 3D dMRI had a low temporal resolution and a long postprocessing time compared to 2D dMRI [18].

#### 4.2.2. Contrast Enhancement-MRI

For pleural lesion evaluation, intravenous contrast administration is mandatory. For this purpose, a 3D gradient echo with fat suppression sequence is suggested with an initial basal acquisition and subsequent post-contrast study [17,22]. Pleural neoplastic lesions, in particular MPM, have a high signal intensity after contrast injection, especially in the late phase. Therefore, a post-contrast study should last at least 5 min [15,17]. Three different sequences commonly used for DCE-MRI studies were evaluated in a study by Ng et al., (2020); the authors compared radial stack-of-stars DCE-MRI, the standard cartesian-based gradient-recalled echo sequence (fast low angle shot, FLASH) and the view-sharing time-resolved imaging with stochastic trajectories methods (TWIST). Radial DCE-MRI showed optimal motion robustness in thoracic imaging with a minimum decrease in signal-to-noise ratio (SNR) compared to FLASH and TWIST sequences [17].

To differentiate between malignant and benign pleural lesions, a novel, semi-objective biomarker, the early contrast enhancement (ECE), was proposed in a study by Tsim et al. [23]. Sixty-six patients with suspect MPM were enrolled in the study. After contrast, injection images were acquired at 40 s, 80 s, 4.5 min, 9 min and 13.5 min. For each different lesion, 5 (in macronodular disease) to 15 (in non-nodular disease) ROIs were identified. The presence of lesion enhancement within 4.5 min was defined by the authors as ECE. Lesions without ECE were defined as benign. Lesions with malignant morphological features were confirmed as malignant despite the presence/absence of ECE, while lesions with benign morphological features associated with ECE were also classified as malignant. ECE evaluation as a potential biomarker showed high inter-observer agreement, with good sensitivity and negative predictive value compared to the simple morphological evaluation provided by CT and MRI without perfusion analysis [23].

Tumor response assessment is another critical issue. Modified Response Evaluation Criteria in Solid Tumors (mRECIST) are usually applied, measuring lesion thickness. CT is still the first line imaging method for MPM mRECIST application [19], but MRI could be more sensitive for this purpose. In 2008, Plathow et al. compared the application of conventional RECIST and mRECIST criteria on both MRI and CT exams, assessing that mRECIST criteria application on MRI could help in the evaluation of early therapeutic response [19] (Figure 3).

The evaluation of tumor vascularization, especially with the development of antiangiogenic drugs, has an important prognostic role. DCE analysis may provide information of lesion response to therapy when morphological evaluation is failing. A quantitative analysis of tumor enhancement was proposed by Tomšič et al., in 2019. DCE parameters were evaluated using two different kinetic models: the extended Tofts model and the adiabatic approximation of tissue homogeneity model [15].

#### 4.2.3. Diffusion-Weighted Imaging and Apparent Diffusion Coefficient

DWI is a type of functional imaging based on the random movements of water molecules in tissues. Many pathologies, such as malignancies and inflammatory disease, can alter water diffusion causing a restriction of molecules movements because of molecular and structural changes [32]. Through post-processing algorithms, it is possible to represent a quantitative analysis of data obtained through DWI as apparent diffusion coefficient (ADC) maps [32]. DWI and ADC maps are currently performed during MRI examinations in many oncological settings, including chest evaluation and assessment [33].

##### ADC Cut-Off

Many studies have been conducted to confirm that both DWI and ADC maps can be used in the evaluation of MPM or malignant pleural disease (MPD) and in differentiating benign from malignant disease. Moreover, a mean ADC value that could reliably be used for distinguishing malignant from benign lesions has been investigated too.

In 2012, Coolen et al. prospectively evaluated the use of DWI in the differentiation of MPD from benign lesions [22]. With a 3T whole-body system, DWI was performed with multiple b values (0, 50, 100, 500, 750 and 1000 s/mm^2^), followed by DCE imaging acquisitions, used retrospectively for the evaluation of misclassified lesions. A statistically significant difference between ADC values of MPD and that of benign lesion was obtained; a mean ADC value of 1.52 × 10^−3^ mm^2^/s was suggested as an optimal cut off (71.4% sensitivity, 100% specificity, 87.1% accuracy, 81% NPV and 100% PPV). Using an ADC range between 1.52 and 2.00 × 10^−3^ mm^2^/s a misclassification of lesions may occur; the authors hypothesized as the presence of necrosis or inflammation in the tumor as a probable explanation. The use of DCE in addition to information gained through ADCs improved the sensitivity to 92.8% with, regardless, a decrease in specificity to 94.1%. Accuracy, PPV and NPV all changed to 93.5%, 92.8% and 94.1%, respectively [22].

In accordance with these results are those of İnan et al., (2016); in this work the authors tried to differentiate metastatic malignant lesions from benign pleural thickening through DWI. Multiple b values were used to acquire diffusion-weighted images (0, 650 and 1000 s/mm^2^) and multiple ADC maps were obtained (ADC_1_ calculated from b value of 0 and 650 s/mm^2^, ADC_2_ obtained from 0 and 1000 s/mm^2^). A statistically significant difference of ADC_1_ and ADC_2_ between metastatic pleural thickening and benign lesions was found (*p* < 0.05). The authors found that a mean ADC value ≤ 1.5 × 10^−3^ mm^2^/s suggests a malignant etiology, despite low sensitivity and specificity [27].

More recently, Usuda et al., in 2019, evaluated DWI and ADC maps to differentiate MPM from other pleural pathologies such as metastases from lung cancer, empyema and pleural effusions [31]. DWI was performed with b values of 0 and 800 s/mm^2^ through a 1.5T superconducting magnetic scanner after anatomical T1w and T2w acquisitions. The mean ADC value obtained for MPM was not significantly different from that of pleural dissemination, whereas a significant difference was found between the ADC values of MPM and those of empyema and pleural effusion, and the same was reported for metastatic lesions. Both MPM and metastatic lesions presented a significantly lower ADC value than that of pleural effusion and empyema [31].

The results obtained by Sabri et al., in 2021 are similar [26]. A 1.5T scanner was used to acquire diffusion-weighted images with three different b values of 0, 500 and 1000 s/mm^2^. A qualitative assessment of the obtained images and a quantitative one with ADC maps were made. All lesions that presented diffusion restriction were proven to be histopathological malignant; conversely, all non-restricted lesions appeared to be benign. The mean ADC value of MPD was significantly lower than that of benign lesions, while no significant difference was found between the ADC value of MPM compared to that of metastatic pleural lesions, as in accordance with Usuda et al. [26,31]. Authors suggested a mean ADC cutoff value of 1.68 × 10^−3^ mm^2^/s as capable of differentiating malignant from benign pleural lesions (*p* < 0.001) [26].

A different result was reported by Jiang et al. in 2021; retrospectively analyzing the diagnostic performance of DWI using a 1.5T superconducting magnet in patients suspected of having pleural malignancies, the author found out that although hyperintense pleural areas on DWI are suggestive of malignancy, especially with high b values, the mean ADC value in the group of pleural malignancies was not statistically different from that of the benign group, nor they could find a cutoff value that was able to discriminate malignant lesions from benign ones. The authors supposed that these results may be the consequence of the impossibility to obtain the ADC value in a large group of benign lesions and that a subjective position of the ROIs chosen for ADC evaluation may have led to bias [30].

Thus, a mean ADC value able to discriminate reliably MPD from benign lesions is still debated and more studies are needed to assess a robust cutoff value (Table 2).

##### ADC-ROIs

The problem of different ROIs positioning in the ADC measurements of pleural abnormalities related to intra- and interobserver variability was assessed by Priola et al. in 2017 [29]. Five different methods of measurement were proposed. Three were based on a manual position of the ROI comprehensive of the entire circumference: Whole Tumor Volume, Three Slice Observer Defined and Single Slice; the other two consisted of the positioning of one (One Small Round ROI) or more than one (multiple small round ROI) small circular ROI in the more restricted area. Each method presented a good to excellent intra- and interobserver concordance, but better results were obtained when the entire tumor in one or more slices was considered, especially with the Single Slice method [29].

##### ADC-Histologic Subtype

The capability to differentiate histologic subtypes of MPM without using an invasive approach was proposed by Gill et al., in 2010 [28].

Through a 3T magnetic resonance, free-breathing DWI was acquired with different b values (250, 500, and 750 s/mm^2^). The final group consisted of 50 investigated patients affected with histologically proven MPM (35 epithelioid, 10 biphasic and 5 sarcomatoid). Each different histologic subtype showed different average ADC values: for the epithelioid it was 1.31 ± 0.15 × 10^−3^ mm^2^/s, for the biphasic it was 1.01 ± 0.11 × 10^−3^ mm^2^/s, and for the sarcomatoid it was 0.99 ± 0.07 × 10^−3^ mm^2^/s. No statistically significant difference was found between ADC values of biphasic and sarcomatoid tumors, while the ADC of epithelioid MPM appeared to be significantly higher than those of biphasic and sarcomatoid MPM. ADC value of the epithelioid subtype showed a sensitivity of 60%, a specificity of 94% and an accuracy of 84% in the differentiation from the sarcomatoid subtype [28].

##### DWI-Visual Assessment

For a more clinical and rapid assessment of pleural lesions, in 2014, Coolen et al., suggested a new visual evaluation of MPD through DWI in a group of patients suspected of being affected by malignant pleural mesothelioma [25]. The analysis of multiple b values (0, 50, 100, 500, 750 and 1000 s/mm^2^) led them to coin the term “pleural pointillism” to report the appearance of multiple hyperintense spots mostly at high values of b parameter (1000 s/mm^2^) as result of MPD in visual imaging examination. This diagnostic marker presented higher values of sensitivity, specificity, PPV, NPV and accuracy (92.5%, 78.8%, 89.9%, 83.9% and 88%, respectively) compared to those obtained at the CT evaluation of mediastinal thickening and shrinking lung appearance. The authors suggested that this pattern of visualization may help in pleural biopsy and limit samples [25].

In accordance with these results are those reported in 2021 by Jiang et al. The authors found areas of signal hyperintensity, with b values of 800 s/mm^2^ and resembling those of pleural pointillism, in 94% of patients with malignant pleural lesions [30].

A different type of appearance of MPM in DWI was reported by Usuda et al. [31]. The authors subclassified different patterns of diffusivity restriction observed in pleural assessment into four categories (strong continuous diffusion, strong scattered diffusion, weak continuous diffusion, and no decreased diffusion). All MPM evaluated presented a strong continuous diffusion pattern on DWI, while a strong scattered pattern was observed mainly in pleural dissemination. No decrease in diffusion was observed in pleural effusion, while the weak continuous pattern was characteristic of empyema [31].

## 5. MRI Protocol: A Proposal

On the basis of the papers found in literature, an MR protocol for patients with MPM was hypothesized and elaborated. The protocol was developed using a 1.5T MRI system (MAGNETOM Symphony, Tim System, Siemens). An eight-channel phased array coil was used, and the patient was in supine position.

We elaborated an MRI scan protocol with pre-contrast acquisitions, including functional scans (DWI) and post-contrast acquisitions, performed at different acquisition times. We used commercially available sequences, while experimental MRI techniques were not considered, letting their use remain limited to a research setting. The proposed MRI protocol for MPM patients is summarized in Table 3.

For a morphological analysis, we proposed a 2D T2-weighted acquisition (Half-Fourier acquisition single-shot turbo spin-echo, HASTE, with and without fat suppression) on an axial, coronal and sagittal plan. Sequences were acquired at end expiration during multiple short breath-hold. A T1-weighted turbo spin echo (TSE) sequence was made to complete the morphological assessment.

For functional imaging, axial DWI with multiple b-value (b = 0, 50, 400, 800 s/mm^2^) was performed to assess the presence of areas of signal restriction (Figure 4).

Finally, T1-weighted fat-saturated 3D gradient echo (VIBE) sequences before and after contrast injection were made. After the administration of contrast injection (0.2 mL/kg of gadoteridol (Prohance^®^; Bracco, Milan, Italy), acquisitions occurred at 40 s, 80 s, 3 min, 4:30 min and at 5 min, to assess lesions enhancement.

Post-processing analysis of the data was also performed with a commercially available software, generating ADC maps.

## 6. Conclusions

MRI application for chest disease evaluation has long been discussed. In this study we revised the potential role of MRI in the assessment of MPD and in the differential diagnosis between malignant and benign lesions. An MRI protocol in order to evaluate MPD with a special focus on MPM was proposed.

Further studies are needed to confirm and improve MRI application for pleural and general chest disease assessment.

## Figures and Tables

**Figure 1 diagnostics-12-00750-f001:**
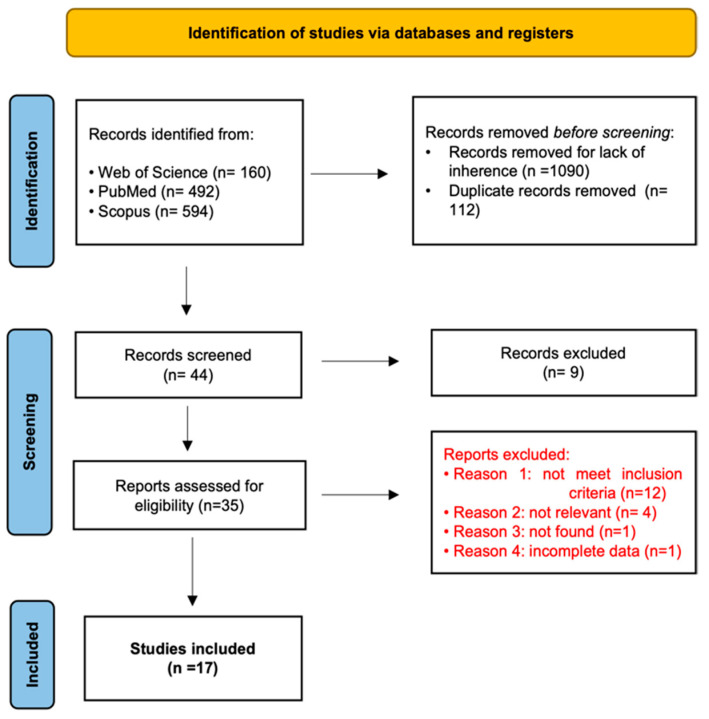
PRISMA flow diagram describing the study selection [21].

**Figure 2 diagnostics-12-00750-f002:**
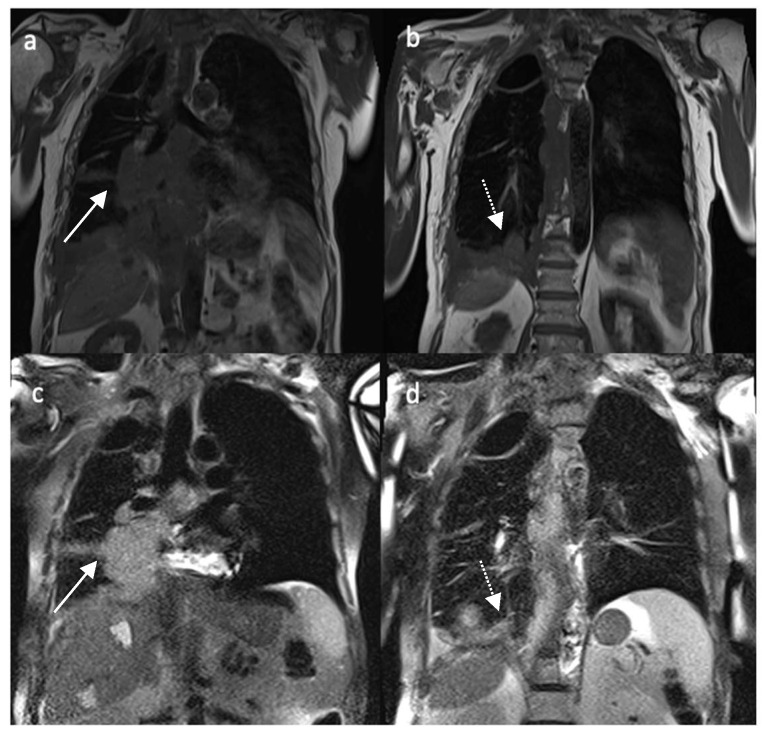
Non-enhanced coronal T1w (**a**,**b**) and T2w (**c**,**d**) images of a patient affected by MPM. White arrow in (**a**,**c**) depicts paramediastinic nodular lesions; dotted white arrow in (**b**,**d**) portrays diaphragmatic pleural thickening.

**Figure 3 diagnostics-12-00750-f003:**
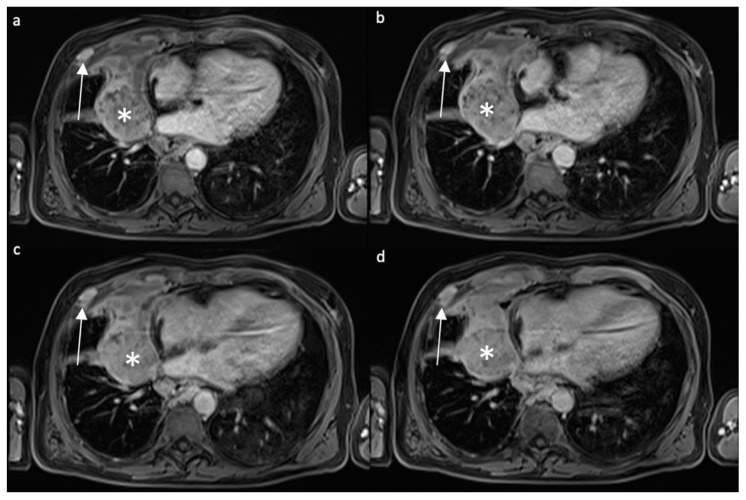
Contrast-enhanced T1w axial fat-saturated images acquired with different timing after contrast administration ((**a**) 40 s, (**b**) 80 s, (**c**) 3′, (**d**) 5′) in a patient affected by MPM. White asterisk: paramediastinal mass with peripheral enhancement, in particular in the anterior portion; white arrow: enhancing nodule in the anterior thoracic wall is present.

**Figure 4 diagnostics-12-00750-f004:**
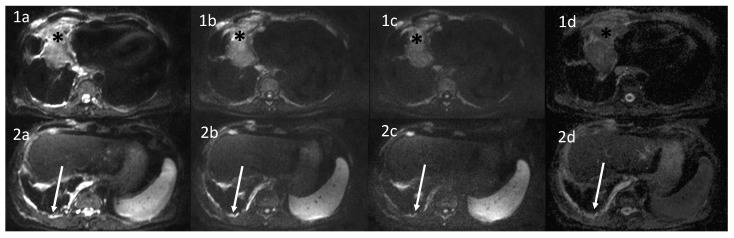
Diffusion Weighted Images (DWI) and Apparent Diffusion Coefficient (ADC) maps of a patient affected by MPM. Different b values were performed (**a**: **b** = 50 s/mm^2^; **b**: **b** = 400 s/mm^2^; **c**: 800 s/mm^2^; **d**: ADC map) to assess persistent areas of signal restriction. (**1a**–**1d**) Black asterisk: previously treated malignant pleural mesothelioma (MPM) with persistence of diffusion restriction in the anterior portion of the lesion. (**2a**–**2d**) Diaphragmatic and mediastinal pleural thickening with evidence of pointillism on DWI.

**Table 1 diagnostics-12-00750-t001:** Description of studies included. MR: Magnetic Resonance; CT: Computed Tomography; PET: Positron Emission Tomography; T1w: T1-weighted; T2w: T2-weighted; SE: spin-echo; DWI: diffusion weighted imaging; FSE: fast spin-echo; GE: gradient-echo; FS: fat saturation; DCE: dynamic contrast enhancement; SPGR: spoiled gradient recalled; FFE: fast field-echo; TSE: turbo spin-echo. SS-TSE: single shot turbo spin-echo. HASTE: half-Fourier acquired single-shot turbo spin-echo; VIBE: Volumetric interpolated breath-hold examination; FLASH: Fast low angle shot; SPIR: spectral pre-saturation with inversion recovery; TWIST: time-resolved with interleaved stochastic trajectories; FISP: fast imaging with steady-state precession.

Publication	Study Design	Study Population (Period, Location)	N Eligible Patients	N Included Patients	N MPD Patients	N MPM Patients	Imaging Technique	MRI Sequences
Podobnik J. et al., 2010 [9]	Prospective study		15	15	10	10	3T MR and CT	Coronal, axial and sagittal T2w TSE with SPIR; axial T1w TSE black blood
Tomšič M.V. et al., 2019 [15]	Prospective study	October 2013–July 2015	29	19	19	19	3T MR	Axial T2w TSE FS; VIBE; DCE turbo-FLASH
Weber M.A. et al., 2004 [16]	Prospective study		21	21		4	1.5T MR and CT	T2w TSE; T1w TSE before contrast and T1w TSE FS TSE after contrast; radial before and after contrast
Ng S. C. et al., 2020 [17]	Prospective study	May 2008–May 2017	23	23			3T and 1.5T MR	3T: axial and coronal HASTE; coronal VIBE; axial DWI; radial DCE; FLASH DCE; TWIST DCE 1.5T: radial DCE
Plathow C. et al., 2006 [18]	Prospective study		22	22	22	22	1.5T MR	trueFISP; FLASH 3D
Plathow C. et al., 2008 [19]	Prospective study		50	50	50	50	1.5T MR and CT	Coronal and axial HASTE; coronal and axial pre and postcontrast VIBE; coronal T2-TSE
Knuuttila A. et al., 2001 [20]	Prospective study	January 1997–December 1998	34	34	27	18	1.5T MR and CT	Precontrast: Axial T1w FLASH; axial T2w true FISP; axial T2w FS HASTE Postcontrast: axial, coronal and sagittal T1w FS FLASH
Coolen J. et al., 2012 [22]	Prospective study	November 2009–May 2010	31	31	14	12	3T MR and PET/CT	Axial T2w SS-TSE; DWI; DCE T1w 3D FFE
Tsim S. et al., 2018 [23]	Prospective study	Not reported	66	58	36	31	3T MR and CT	Pre- and postcontrast coronal T1w FS 3D SPGR
Hierholzer J. et al., 2000 [24]	Retrospective study	January 1992–June 1998	88	42	27	9	1.5T MR and CT	T1w pre- and postcontrast; T2w TSE
Coolen J. et al., 2015 [25]	Prospective study	November 2009–December 2012	109	100	67	57	3T MR, CT and PET/CT	T2w SS-TSE FS; DWI
Sabri Y. et al., 2021 [26]	Prospective study	March 2019–November 2020	57	57	28	7	1.5T MR	Axial and coronal T1w TSE; axial, coronal and sagittal T2w TSE; axial STIR; axial DWI
Inan N. et al., 2016 [27]	Prospective study	November 2013–September 2014	42	34	19	0	3T MR	Axial T1w SPGR-FFE with and without FS; coronal and axial T2w SS-TSE; axial T2w SS-TSE with FS; DWI
Gill R. R. et al., 2010 [28]	Prospective study	June 2008–January 2009	62	62	59	57	3T MR	Coronal and axial HASTE; 3D T1w GE; DWI
Priola A.M. et al., 2017 [29]	Retrospective study	January 2014–July 2016	37	34	34	18	1.5T MR	Axial, coronal and sagittal DWI; T2w SS-TSE; T1w fast field echo
Jiang W. et al., 2021 [30]	Retrospective study	March 2014–August 2018	730	70	52	1	1.5T MR and CT	Axial T1w; axial T2w; DWI
Usuda K. et al., 2019 [31]	Prospective study	March 2015–February 2019	43	43	21	11	1.5T MR, CT, PET/CT	Coronal T1w SE; coronal and axial T2w FSE; DWI

**Table 2 diagnostics-12-00750-t002:** Main paper with relative apparent diffusion coefficient (ADC). MPM: malignant pleural mesothelioma. MPD: malignant pleural disease.

Publication	Lesion	Mean ADC Value	Notes
Coolen J. et al., 2012 [25]	Malignant Pleural Disease Benign Alterations	1.40 ± 0.33 × 10^−3^ mm^2^/s 2.49 ± 0.81 × 10^−3^ mm^2^/s	ADC MPD vs. benign alterations (*p* < 0.001)
Sabri Y. et al., 2021 [26]	Malignant Pleural Lesions Benign Pleural Lesions MPM Pleural Metastases	1.10 ± 0.53 × 10^−3^ mm^2^/s 2.19 ± 0.42 × 10^−3^ mm^2^/s 0.84 ± 0.22 × 10^−3^ mm^2^/s 1.19 ± 0.58 × 10^−3^ mm^2^/s	ADC malignant vs. benign pleural lesions (*p* < 0.001) ADC MPM vs. Pleural Metastases (*p* = 0.090)
İnan N. et al., 2016 [27]	Metastatic Malignant Pleural Thickening Benign Pleural Thickening	1.37 ± 0.65 × 10^−3^ mm^2^/s (ADC_1_) 1.06 ± 0.56 × 10^−3^ mm^2^/s (ADC_2_) 2.11 ± 0.69 × 10^−3^ mm^2^/s (ADC_1_) 1.56 ± 0.71 × 10^−3^ mm^2^/s (ADC_2_)	ADC_1_ and ADC_2_ of Metastatic Malignant Pleural Disease vs. Benign Disease (*p* < 0.05)
Gill R. R. et al., 2010 [28]	Epithelioid MPM Biphasic MPM Sarcomatoid MPM	1.31 ± 0.15 × 10^−3^ mm^2^/s 1.01 ± 0.11 × 10^−3^ mm^2^/s 0.99 ± 0.07 × 10^−3^ mm^2^/s	ADC Epithelioid vs. Sarcomatoid (*p* < 0.05) ADC Epithelioid vs. Biphasic (*p* < 0.05)
Jiang W. et al., 2021 [30]	Malignant Group Benign Group	1.15 ± 0.32 × 10^−3^ mm^2^/s 1.46 ± 0.68 × 10^−3^ mm^2^/s	ADC Malignant vs. Benign Group (*p* = 0.161)
Usuda K. et al., 2019 [31]	Pleural dissemination MPM Empyema Pleural Effusion	1.31 ± 0.49 × 10^−3^ mm^2^/s 1.22 ± 0.25 × 10^−3^ mm^2^/s 2.01 ± 0.45 × 10^−3^ mm^2^/s 3.76 ± 0.62 × 10^−3^ mm^2^/s	ADC MPM vs. Empyema (*p* = 0.0007) ADC MPM vs. Pleural Effusion (*p* < 0.0001) ADC of MPM vs. Pleural Dissemination: not significantly different

**Table 3 diagnostics-12-00750-t003:** Proposed MRI protocol for patients with malignant pleural mesothelioma. TSE: turbo spin echo; HASTE: half-Fourier acquired single shot turbo spin-echo; VIBE: Volumetric interpolated breath-hold examination; DWI: diffusion weighted imaging; SPAIR: spectral attenuated inversion recovery; FOV: field of view; TR: repetition time; TE: echo time; FA: flip angle.

Sequence	Manufacturer Acronyms	Typical Contrast	Average Acquisition Time	Spatial Resolution Scan Plane	Scan Parameters	Field Strength B0
Morphology						
2D Echo-planar Fast Spin Echo Sequence	HASTE (SIEMENS)	T2-weighted	Expiration Breath-Hold ≈ 40 s (in 2 different breath-hold)	FOV = 440 mm Thickness = 5 mm Plane = Coronal	TR = 2860 ms TE = 93 ms FA = 160 deg Matrix = (182 × 256)	1.5T
2D Echo-planar Fast Spin Echo Sequence	HASTE (SIEMENS)	T2-weighted	Expiration Breath-Hold ≈ 40 s (in 2 different breath-hold)	FOV = 420 mm Thickness = 5 mm Plane = Sagittal	TR = 2860 ms TE = 93 ms FA = 160 deg Matrix = (182 × 256)	1.5T
2D Echo-planar Fast Spin Echo Sequence	HASTE (SIEMENS)	T2-weighted	Expiration Breath-Hold ≈ 40 s (in 2 different breath-hold)	FOV = 360 mm Thickness = 5 mm Plane = Axial	TR = 2860 ms TE = 93 ms FA = 160 deg Matrix = (170 × 256)	1.5T
2D Echo-planar Fast Spin Echo Sequence (Fat Saturated)	HASTE (SIEMENS)	T2-weighted (Fat saturated)	Expiration Breath-Hold ≈ 40 s (in 2 different breath-hold)	FOV = 440 mm Thickness = 5 mm Plane = Coronal	TR = 2860 ms TE = 93 ms FA = 160 deg Matrix = (182 × 256)	1.5T
2D Echo-planar Fast Spin Echo Sequence (Fat Saturated)	HASTE (SIEMENS)	T2-weighted (Fat saturated)	Expiration Breath-Hold ≈ 40 s (in 2 different breath-hold)	FOV = 440 mm Thickness = 5 mm Plane = Axial	TR = 2860 ms TE = 93 ms FA = 160 deg Matrix= (170 × 256)	1.5T
2D Turbo Spin Echo (TSE)		T1-weighted	Free breathing (Navigator) ≈ 2:30 min	FOV = 440 mm Thickness = 5 mm Plane = Coronal	TE =491 ms TE = 19 ms FA = 135 deg Matrix = (179 × 256)	1.5T
3D Rapid Acquisition Spoiled Gradient Echo	VIBE (SIEMENS)	T1-weighted (Fat saturated)	Expiration Breath-Hold ≈ 26 s	FOV = 390 mm Thickness = 2.5 mm Plane = Axial	TR = 4.55 ms TE = 2 ms FA = 10 deg Matrix = (153 × 224)	1.5T
DWI						
2D Single-Shot Echo Planar Imaging (EPI)	DWI	T2-weighted diffusion-weighted (SPAIR fat saturation)	Free breathing (Navigator) 7–10 min	FOV = 360 mm Thickness = 5 mm Plane = Axial	TR = 11,500 TE = 90 BW = 1371 B = 0, 50, 400, 800 s/mm^2^ Matrix = (148 × 192)	1.5T
Contrast Gadolinium Enhanced						
3D Rapid Acquisition Spoiled Gradient Echo Post-contrast (40 s, 80 s, 3′, 4:30′, 5′)	VIBE (SIEMENS)	T1-weighted (Fat saturated)	Expiration Breath-Hold ≈ 26 s	FOV = 390 mm Thickness = 2.5 mm Plane = Axial	TR = 4.55 ms TE = 2 ms FA = 10 deg Matrix = (153 × 224)	1.5T

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
