# Peer review of "The Use of Chest Magnetic Resonance Imaging in Malignant Pleural Mesothelioma Diagnosis"

_diagnostics, 2022, doi:10.3390/diagnostics12030750_

Round 1

Reviewer 1 Report

The review presented by F. Volpi and colleagues is of great interest, showing many improvements in MRI techniques over the last two decades that have contributed to a better and early detection of malignant pleural mesothelioma in patients. The differentiation between malignant and benign lesions, analyzed in the various chapters / paragraphs would be of high value to readers as this point has not really been well documented in the literature until now.

The manuscript is clear, overall well presented, and the method used fine, however I think some important improvements could be made according to the points listed below:

1) The authors did not consider two main references relevant to the topic, which could be (at least) be mentioned in the introduction. These correspond to two reviews of the 11th and 12th International Conference of the International Mesothelioma Interest Group (IMIG), published in 2013 and 2016, respectively. The first one is important as it presented a review of pointillism on DWI, a potential predictor of MPM and also useful for avoiding unnecessary invasive procedures... The second one gave some additional advantages of MRI vs CT, etc... Samuel G. Armato III, et al. Imaging in pleural mesothelioma... Lung Cancer 82 (2013), 190-196; Samuel G. Armato III, et al. Imaging in pleural mesotheioma... Lung Cancer 101 (2016), 48-58.

2) Another work of interest was the report of Katharina Martini et al. Diagnostic accuracy of sequential... Lung Cancer (2016) 40-45, regarding the diagnostic accuracy of PET + MR, in comparison with PET/CT.

3) Finally, the work of Yoshiharu Ohno at al. (Whole-body MRI..., AJR (2019) 212, 311-319) is important too to my opinion, and relevant to this review. Please see whether it could be taken into consideration and where in the manuscript.

4) One last suggestion: may be the authors could mention the fact that the number of reviews published in this field has doubled in the years 2012-2021 compared with the previous decade 2002-2011, as an evidence of the growing interest of this topic.

5) Table 3 is not very clear, please check the resolution.

Reviewer 2 Report

The authors provide a thorough and objective summary of the application of MR based imaging for the diagnosis and characterization of MPM. Furthermore, this information is distilled into a proposed protocol for others to apply at their respective institute. The data is well presented and aided with clear figures to illustrate these concepts. I have only a minor suggestion.

Line 131 – Wording is confusing, consider revising – “usually a first morphological study”
